# A journey to a new stable state—further development of the postoperative recovery concept from day surgical perspective: a qualitative study

Ulrica Nilsson  ,[1,2] Maria Jaensson,[3] Karin Hugelius,[3] Erebouni Arakelian,[4] Karuna Dahlberg[3]

¹Department of Neurobiology, Care Sciences and Society, Karolinska Institutet, Stockholm, Sweden
²Perioperative Medicine and Intensive Care, Karolinska Universitetssjukhuset, Stockholm, Sweden
³School of Health Sciences, Faculty of Health and Medical Sciences, Örebro University, Örebro, Sweden
⁴Department of Surgical Sciences, Uppsala University, Uppsala, Sweden

**Correspondence to**
Professor Ulrica Nilsson;
ulrica.nilsson@ki.se

## ABSTRACT

**Objective** This study aims to further develop the concept analysis by Allvin et al in 2007 and Lundmark et al in 2016 from the perspective of day-surgery patients. Also, to describe how patients experience postoperative recovery in relation to the identified dimensions and subdimensions and to interpret the findings in order to get a deeper understanding of the concept postoperative recovery.

**Design** Descriptive qualitative design with a theoretical thematic analysis.

**Setting** Six day-surgery departments in Sweden.

**Participants** Thirty-eight adult participants who had undergone day surgery in Sweden. Participants were purposively selected.

**Results** Four dimensions—physical, psychological, social and habitual—were confirmed. A total of eight subdimensions were also confirmed, two from Allvin et al's study and six from Lundmark et al's study. Recovery included physical symptoms and challenges coping with and regaining control over symptoms and bodily functions. Both positive and negative emotions were present, and strategies on how to handle emotions and achieve well-being were established. Patients became dependent on others. They coped with and adapted to the recovery process and gradually stabilised, reaching a new stable state.

**Conclusion** Postoperative recovery was described as a process with a clear starting point, and as a dynamic and individual process leading to an experience of a new stable state. The recovery process included physical symptoms, emotions and social and habitual consequences that challenges them. To follow-up and measure all four dimensions of postoperative recovery in order to support and understand the process of postoperative recovery is, therefore, recommended.

## INTRODUCTION

Over time, there has been a transition from inpatient surgery to outpatient surgery— that is, day surgery. Day surgery is defined as surgery in which the surgical patient is admitted and discharged on the same day, or within 24 hours,[1] and accounts for 70%–75% of all surgeries performed.[2] However, the

### Strengths and limitations of this study

► The concept of postoperative recovery has been described earlier yet not in context of day-surgery patients.
► This qualitative study was performed on a relatively large sample.
► All researchers have theoretical and practical knowledge of the postoperative context.
► The interviews were conducted 14–57 days after the surgery, which implies that some of patients had not recovered yet.

rates of specific types of day surgery, such as carpal tunnel release and cataract extraction, are higher than 90%.[3] Day surgery is preferred by both healthcare providers and patients.[1 4 5] It provides quick and effective care with minimal interruption of patients' daily life, and most patients prefer recovering at home over staying at the hospital.[6] However, day surgery requires patients to take greater responsibility for their recovery process.[7 8] Although there is no standard definition of postoperative recovery, it is commonly known as 'an energy-requiring process of returning to normality and wholeness'.[9] This statement raises several questions: for example, should postoperative recovery be seen as an endpoint or as a process,[10] and what does it mean to be 'recovered'?[11]

Postoperative recovery can be described in three phases. Phase I (early phase) starts when the patient leaves the operating room. For medical care, this phase includes close monitoring of vital parameters. Phase II (intermediate phase) begins when the patient is still cared for at the hospital but is not monitored as closely as in phase I.[12] These two phases focus on goals such as loss of pain, regaining reflexes, and loss of postoperative nausea and vomiting.[11] Patients stay for approximately

1–2 hours at the hospital after day surgery.[13 14] Phase III (late phase) occurs when the patient is discharged from hospital.[9 11 12] Phase III recovery takes time, and has been defined by Alvin *et al*[9] as a process of complete return to the usual self or to preoperative health status (or better). It can be a complex and fragile process, with physical, emotional, social and habitual characteristics.[9] For patients to play an active role in their own recovery process, ongoing support from healthcare and next of kin is required after discharge.[7 8 15] Therefore, it is important to understand what the recovery process at home includes and how it can be defined.

Alvin *et al*[9] used Walker and Avant's concept analysis approach to define the concept of postoperative recovery, based on four identified dimensions of postoperative recovery: physiological, psychological, social and habitual recovery. In 2016, Lundmark *et al*[16] further developed this concept from lung recipients' perspective of their post-transplant recovery process. They found that Allvin's concept analysis was partly applicable to the context of lung transplantation. The main dimensions of the concept analysis were confirmed, although several subdimensions were found to be contradictory and were excluded. Six new subdimensions emerged: symptom management, adjusting to physical restraints, achieving an optimum level of psychological well-being, emotional transition, social adaptation and reconstructing daily occupation.[16] Alvin *et al*'s work is based on rather the old literature, published from 1982 to 2005, and includes both inpatients and outpatients,[9] while Lundmark *et al*'s concept was a further development of Alvin *et al*'s work in order to suit lung transplant recipients.[16] Consequently, there is a need to further develop the concept of postoperative recovery after day surgery to obtain a deeper understanding. Several studies report that day-surgery patients feel lonely and notice a gap between care provided at the hospital and care they receive after discharge.[7 8 17 18] It is, therefore, imperative to examine that gap and obtain a better understanding of patients' perspective on recovery after day-surgery procedures. Since the number of patients undergoing day surgery is increasing, deeper understanding of the concept of postoperative recovery is necessary in this patient group, in order to better understand the concept of postoperative recovery described by Alvin *et al*[9] and reanalysed by Lundmark *et al*.[16] Otherwise, there is a risk that this concept will be used in a population that it is not developed for,in this case, day-surgery patients. This study focuses on the late postoperative recovery process, which occurs after discharge, when the patient is left alone without monitoring by healthcare.

## THE STUDY
### Aims
The aims of this study are threefold:
► To further develop the concept analysis by Alvin *et al*[9] and Lundmark *et al*[16] from the perspective of day-surgery patients.

► To describe how patients experience postoperative recovery in relation to the identified dimensions and subdimensions.
► To interpret the findings in order to get a deeper understanding of the concept postoperative recovery.

### Design
This qualitative study has a descriptive design that uses Alvin's[9] and Lundmark's[16] previous descriptions of postoperative recovery as a basis for theoretical thematic analysis.[19] Material from two previous studies was reanalysed: sample I[8] and sample II.[18] Both studies are presented elsewhere.[9 16] The Standards for Reporting Qualitative Research reporting guidelines[20] has been followed.

### Participants
The sample consists of 38 participants in total (sample I, n=18 and sample II, n=20), who underwent a day-surgical procedure. Inclusion criteria in both samples were: undergoing day surgery, being >17 years of age, being able to understand written and spoken Swedish. Sample I also included the inclusion criteria: having access to a smartphone and 14 days of digital postoperative follow.[8] Exclusion criteria in both samples were: visual impairment, memory impairment, substance abuse or undergoing a surgical abortion. In both samples' participants were consecutively purposively selected, with the aim to include a maximum variation regarding age, gender, anaesthesia and type of surgery. Participants received study information and invitation to participate by two of the authors (KD in sample I and KH in sample II).

### Patient and public involvement
No patient was involved in the design of this research. The participants, as described below, were involved in this research through interviews where they contributed with their experiences about their postoperative recovery.

### Data collection
The 18 participants in sample I underwent surgery between December 2015 and July 2016, and were recruited from 4 different day-surgery departments.[8] The remaining 20 participants (sample II) underwent surgery from June to September 2017 and came from 2 different day-surgery departments[18] (table 1). Interviews were performed by two of the authors (KD for sample I and KH for sample II). Interview locations were chosen by the participants (table 2). All interviews were audio recorded and transcribed verbatim.

The interview guides used for both samples I and II consisted of questions that mainly focused on the participants' experiences of postoperative recovery after discharge (online supplemental file). Thus, patients were asked about their experience of the first day after surgery (samples I and II) and their recovery until the interview date (first 14 days for sample II and first 22–57 days for sample I). To cover patients' experiences of recovery after day surgery, patients were asked to reflect on having the surgery as a day surgery, and to compare their experience

**Table 1** Distribution of sex, age, type of surgery, anaesthesia and time of interview between the two samples

|  | Sample set I n=18 | Sample set II n=20 |
|---|---|---|
| **Sex** |  |  |
| Female, n | 10 | 11 |
| Male, n | 8 | 9 |
| **Age** |  |  |
| Mean (min.–max.) | 49.5 (21–80) | 49 (18–76) |
| **Type of surgery, n** |  |  |
| General | 5 | 7 |
| Hand | 5 | 9 |
| Orthopaedic | 7 | 4 |
| Ear, nose and throat surgery | 1 | – |
| **Type of anaesthesia, n** |  |  |
| General | 14 | 10 |
| Regional | 4 | 10 |
| **Postoperative day of interviews, mean (min.–max.)** | 36 (22–57) | 14 (12–19) |
| **Media of interview, n** |  |  |
| Face to face |  |  |
| At the participant's home | 7 | 5 |
| At the participant's workplace | 3 | – |
| At the university | 7 | 1 |
| At the hospital | – | 1 |
| Skype | 1 | – |
| Phone | – | 13 |

with any previous experiences of undergoing surgery. No new information occurred in the last interviews conducted in neither sample I and sample II, which suggest that saturation was reached.

## Data analysis

Theoretical thematic data analysis[19] was performed, using the concept analysis by Allvin *et al*[9] and Lundmark *et al*[16] to guide the analysis on dimensions and subdimensions of postoperative recovery in day-surgery patients.

The analysis process was as follows:

1. Transcribed interviews from each sample (I and II) were read separately several times by two of the authors (EA and UN) independently. The other authors (KD, MJ and KH) were familiar with the data, since they had been involved in the original data collections and had read the data several times.
2. Two of the authors (EA and UN) conducted a theoretical analysis of the key components of postoperative recovery based on the work of Allvin *et al*[9] and Lund-

mark *et al*.[16] Thereafter, all five authors processed the analysis.
3. The authors jointly finalised the results by finding descriptions and citations from the interviews that captured the content of each subdimension. The authors then reflected on the findings and discussed different ways of interpreting the results in relation to the subdimensions and the theoretical frame of the concept as presented by Allvin *et al*[9] and Lundmark *et al*.[16]
4. Finally, the authors conducted an interpretation the findings in steps 1 and 2 in order to get a deeper understanding of the concept postoperative recovery.

### Rigour

Credibility was guaranteed by having two authors (EA and UN) conduct the analysis separately, and then discuss it with the remaining authors (KD, KH and MJ), all of whom were familiar with the data corpus, until consensus. To enhance transferability, the data analysis was clearly described in order to allow readers to form their own judgement, as far as possible. Credibility and confirmability also come from all of the authors (EA, UN, KD, KH and MJ) having a theoretical and practical knowledge of the postoperative context, which improved their understanding[21] of the concept of postoperative recovery.

## FINDINGS

Participants' experiences of the postoperative recovery process after day surgery was confirmed to fall within the four main dimensions described by Allvin *et al*[9] and Lundmark *et al*.[16] In total, nine subdimensions were found; of these, eight subdimensions aligned with those identified by Allvin *et al*.[9] (n=2) and Lundmark *et al*.[16] (n=6). One subdimension was changed to the opposite meaning: from 'becoming independent' to 'becoming dependent'. There was also a linguistic change of the name of one of the subdimensions that originated from Allvin *et al*.[9] Table 2 presents the subdimensions identified by Allvin *et al*,[9] Lundmark *et al*[16] and the present study.

### Physical dimension

The physical dimension describes physical symptoms and signs related to the surgery and postoperative recovery process, along with the challenges of coping and regaining control over bodily functions and the physical problems that were experienced.

#### Regaining control over bodily functions

Regaining control over reflexes occurs during hospital recovery; therefore, this formulation was deleted from the original subdimension. The participants described regaining control over bodily functions directly after surgery; as Informant 6 (sample I) described: 'I've got to build up (my muscle strength) from scratch, as I've lost my muscle strength'. In order to regain control and cope with the physical consequences of the surgery, the participants described learning new ways of using their

**Table 2** Development of the concept of postoperative recovery from Allvin et al[9] and Lundmark et al[16] to suit late postoperative recovery in day-surgery patients

| Dimensions | Allvin et al Subdimensions | Lundmark et al Subdimensions | Late recovery after day surgery Subdimensions |
|---|---|---|---|
| Physical | Regaining control over reflexes and motor activities | Regaining control over reflexes and bodily functions | Regaining control over bodily functions |
|  | Normalise and control bodily functions | – | – |
|  | Loss of pain and fatigue | – | – |
|  | Conservation of energy | Conserving energy | Conserving energy |
|  | Experience of passivity | Experiencing passivity | – |
|  |  | Symptom management | Symptom management |
|  |  | Adjusting to physical restraints | – |
| Psychological | Experience of passivity | Experiencing passivity |  |
|  | Return to psychological well-being | – | – |
|  | Return to wholeness | – | – |
|  | Reinstate integrity | Reinstating integrity |  |
|  | Transition from illness to health | Transition from illness to health |  |
|  | Loss of depression, anger, anxiety, fatigue and passivity | – | – |
|  | Experiences of pressures and cues | – | – |
|  |  | Emotional transition | Emotional transition |
|  |  | Achieving an optimum level of psychological well-being | Achieving an optimum level of psychological well-being |
| Social | Becoming independent | Becoming independent | Becoming dependent |
|  | Stabilise at full social function |  | – |
|  | Functioning in interaction with other people |  | – |
|  |  | Social adaption | Social adaption |
| Habitual | Stablising the full range of activities | – | – |
|  | Take responsibility for and controlling activities in daily care | – | – |
|  | Restoration of normal eating, drinking and toilets habits | – | – |
|  | Returning to work and driving | – | – |
|  |  | Reconstructing daily occupation | Reconstructing daily occupation |

body and aids/tools, such as crutches. Despite difficulties controlling bodily functions, the participants felt responsible for contributing to their own recovery, such as by exercising.

> Moving was quite wobbly, it takes a while before you learn how to walk with crutches…The brain has to learn to find a new balance point…. it takes a while… (Informant 4, sample I).

Physiotherapists were important facilitators in the patients' work to regain control over their bodily functions. Physiotherapists' support was needed for patients to challenge themselves to improve their physical functions with exercise, without jeopardising their recovery. Some participants were surprised when they realised that recovering and regaining control over bodily functions took longer than expected. Some commented that they were counting the days until they would feel recovered and achieve a new, stable physical state.

### Conserving energy

Because they felt exhausted, the postoperative patients felt that they had to conserve energy. 'Being exhausted' was described as extreme tiredness, having difficulty

concentrating and being very emotional. In order to cope with daily life, patients needed to conserve their energy, which limited their ability to participate in daily activities. Their body restricted their activities, and they needed to balance activities with rest to conserve energy.

> … to listen to your body … you need to rest, but still be able to walk … to go to the store, take the car and go shop … and then rest. I felt it… to put up my leg and rest … I was operated on a Tuesday, I went to work on Monday … and worked just over half the day, then I just collapsed… (Informant 17, sample I).

The participants described the importance of giving themselves time to rest and move slowly forward, step by step. Being on sick leave due to the surgery and subsequent recovery process was a new experience for some. However, many participants also described their adaptation and adjustment to the demand to conserve their energy, and expressed satisfaction with their general situation.

### Symptom management

The recovery process involved the management of physical symptoms. In general, such postoperative symptoms bothered the participants and caused worries and questions. The participants described symptoms of pain and swelling in the surgical wound or pain in other body parts. Other symptoms were signs of infection, fever, dizziness, bleeding from the wound, difficulty concentrating, numbness in parts of the body, gastroenterological problems and symptoms related to the plaster.

In order to manage their symptoms, participants contacted their healthcare services for advice or information, or used different self-management strategies. Described self-management strategies included positioning the surgical area high up to avoid swelling and pain, eating food with a laxative effect and being observant for signs of wound infection. Participants managed their pain by using different pain management strategies such as avoiding activities or movements causing pain, or using pain medications. When the prescribed pain medication was insufficient, participants tried to solve the problem themselves by asking family members or friends if they had other pain medication. Others chose to avoid taking pain medication because they found that it affected them negatively.

> I needed some help with pain relief for at least a couple of days … but … "No", the nurse said, "it is not possible", … I just had to fight by myself … while at home … (Informant 5, sample II).

### Psychological dimension

The psychological dimension describes positive and negative emotions, strategies to handle emotions and achieve well-being and reflections on the information provided and the recovery process.

### Emotional transition

A range of positive feelings were expressed, such as happiness and gratitude. Some participants described an emotional transition from being very energised and positive immediately after the surgery, to experiencing a setback after this positive period, and described it as being 'wound up … then, some days later, you lose your breath' (Informant 6, sample II). Worries were commonly expressed about the consequences of the surgery, and how these would influence patients' recovery and their ability to return to work.

> What will this mean now for my job … you are a little depressed by it because it hurts all the time … this setback … otherwise, it has been really good … (Informant 1, sample II).

Participants also expressed worry about the results of the surgery, setbacks in their recovery, unexpected pain and what their 'new stable state' would be like. Correctly given and perceived information and support from healthcare were necessary to build realistic expectations. Lack of information—or misinformation—regarding the expected recovery process and the expected 'new stable state' (ie, in terms of normal function, or if normal function could not be expected, and how they would adapt to the new situation) led to worries and frustration.

> How will it [the result of the surgery] go? How will my hand work? … I don't know how to manage my job [since I need my hand]… It is difficult now because the wrist is as bad as it is … I am just a little worried about whether I can work or not … (Informant 7, sample II).

In addition to not receiving information from the surgeon who performed the surgery, patients described receiving information 'too soon' postoperatively, while they were still affected by the anaesthesia. When patients were discharged from the recovery unit before they felt fit to go home, they were left with unanswered questions that caused trouble and worries when they arrived at home.

> He (the surgeon) said I can walk freely. What does it mean? How much should it hurt? Because it's only the pain that sets the limit, and everyone has different pain thresholds, he said … not very clear information … (Informant 6, sample 2).

### Achieving an optimum level of psychological well-being

The participants focused on minor progression in their recovery, and mentioned that their recovery was satisfactory and went better than expected. They felt that they reached an acceptable level of well-being, despite setbacks and problems, such as pain and tiredness. Some participants described how recovering at home and having support from their families and friends were the main factors in achieving an optimum level of psychological well-being during recovery. One strategy to handle psychological well-being was to divide the recovery period

into shorter time periods and plan activities that distracted the participants from their frustration over their symptoms and the time it took before they felt recovered or reached their 'new stable state'.

> We are going away this weekend and that will be nice, you get some, have to try to find things to do even if I'm … when I have this [surgery and issues with the plaster. (Informant 2, sample 1).

### Social dimension

The social dimension comprises participants' reflections on becoming dependent, on limitations due to their surgery and on how they had coped with and adapted to the recovery process and their new stable state.

#### Social adaptation

The participants felt that it was important to continue with their social life and normal activities, both for themselves and for their families. Social adaptation could be facilitated by using digital follow-up support or planning, and by receiving help from family and friends to engage in activities, even when the participant had restrictions. Participants also experienced limitations in social activities, such as attending activities or visiting friends.

> … you can't exercise as usual. You don't meet friends like you usually do, it's a little … a little hard, actually … you don't go and exercise in the same way [as before surgery]" (Informant 5, sample II).

Fear of participating in activities or events where there was a risk that people would bump into them was an obstacle to regaining full social function.

#### Becoming dependent

In contrast to the subdimension 'being independent' by Allvin et al,[9] the participants in the present study described how they became dependent on support from family, friends and healthcare services during their recovery process. For example, when being discharged from the hospital, they needed help with transportation home from their family. They also needed a family member to be vigilant and supportive, and to stay overnight for the first few postoperative days.

> I sat in the back seat on the way home and my parents drove me home … I had great backup from, like, my family and … my husband … (Informant 12, sample I).

Being dependent was experienced as 'being paralysed' and, although it was not always desirable, the participants asked for help from family, friends or neighbours in activities that they had managed themselves before surgery. Feeling insecure if they were sent home too early, not feeling ready to be discharged or other worries made the patients dependent on professional care from the healthcare system. Such needs were expressed by asking for follow-up and support from healthcare after discharge, in order to manage expected and unexpected issues during their recovery period.

> … the instructions you get after such an operation is that the swelling must absolutely not remain because then there can be difficult complications … I wanted to talk to her [the nurse] about it, because it was very swollen … (Informant 2, sample I).

### Habitual dimension

The habitual dimension describes how the participants recovered to their new stable state, as they gradually stabilised their daily activities, their desire to return to work and their worries about working.

#### Reconstructing daily occupation

Participants gradually stabilised daily activities that they had managed to do independently before surgery, such as getting dressed, taking a shower, handling toilet habits and other everyday activities such as taking long walks. Success in common everyday activities such as cooking by themselves was expressed as 'an art'. Personal characteristics such as endurance positively contributed to the reconstruction; as one of the participants commented, you have to be 'somewhat stubborn' (Informant 8, sample I). There were also activities that the participants could not manage yet but yearned to do.

> … it is not easy to take a shower by yourself and not unbutton pants or pull up pants, it is difficult to use four fingers at the moment on one hand …. Well, I'm a little bit slower when I do things, simply … when I make food, it may take half an hour extra … (Informant 2, sample II).

The participants described a will and desire to recover and return to work and ordinary life. Some held to the desire to do activities that had not been possible before the surgery, which were now possible again thanks to the surgery. It was sometimes difficult for patients to accept restrictions and being absent from the work due to a 'small' bodily dysfunction or postoperative health consequences. Therefore, it was very important for them to be able to return to ordinary habitual activities such as driving or biking, since these activities indicated that recovery was in process.

> I really look forward to just having to start driving a car again and start riding a bike…. (Informant 16, sample I).

For some participants, a substitute was not an option; therefore, they declined sick leave and stretched their limits to return to work. At the same time, some participants worried that they would be unable to work independently and would be dependent on colleagues, due to immobility or not having regained enough function or strength after the surgery.

### The process of postoperative recovery

The findings emphasise postoperative recovery as a process, starting from a 'presurgery state' and ending

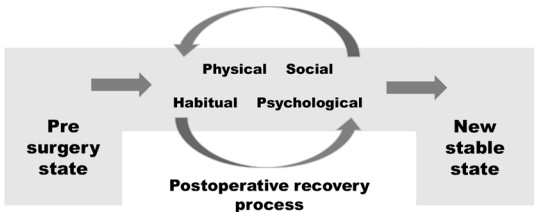

**Figure 1** The process of postoperative recovery.

with a 'new stable state' (figure 1). Postoperative recovery is an individual process and a transformative journey, including how the physical, psychological, social and habitual dimensions affect each other in a continual and dynamic process that leads to a new stable state. This new stable state is not necessarily a state that is the same as that before surgery, or a return to presurgical functions. In many cases, the surgery itself was done to improve mobility and to reduce or remove dysfunctions. For example, some participants described how they were now able to do activities that were impossible before the surgery. For these participants, the new stable state was experienced as being more functional than the preoperative state. In other cases, such as when patients could not return to work, the new stable state involved adjusting to a state with a permanent decrease in or loss of preoperative functions.

## DISCUSSION

This study found that the main dimensions and some of the subdimensions described by Allvin et al[9] and Lundmark et al[16] were applicable to recovery after day surgery. However, recovery after day surgery makes the patient dependent on their surroundings; therefore, one subdimension was linguistically changed. It should be noted that in the present redevelopment of the concept of postoperative recovery, we focused on late postoperative recovery. All interviews in this study were performed between postoperative day 14 and 57, and focused on recovery after discharge. Allvin et al's[9] concept analysis included both early and late postoperative recovery. This difference may explain why the subdimension 'regaining control over bodily functions' in the current study did not include 'control over reflexes', since recovery of protective reflexes and motor activity mainly occurs in the first phase of the recovery process. Another subdimension that differed from the original description by Lundmark et al[16] was 'achieving optimal psychological well-being'.[16] From a day-surgery perspective, this subdimension was described as maintaining a positive attitude, focusing on positive aspects and achievements and the importance of support from the next of kin. Although the original subdimension also described maintaining a positive attitude, the meaning was that patients were grateful for the opportunity for a new chance in life.[16] Another difference was the importance of getting enough information to raise reasonable expectations and support from next of kin and healthcare services. Both Allvin et al[9] and

Lundmark et al[16] briefly mentioned the need for social support after surgery; however, information about what to expect after surgery was not mentioned at all, and the focus was on the positive effect of social support. These aspects may have been emphasised in the present study due to a mismatch between expectations and reality for patients undergoing day surgery. Previous studies have shown that day-surgery patients expect their recovery to be fast and smooth, and to have a minimal impact on their everyday life[6]; similarly, the participants in this study expressed surprise that their recovery was more demanding and took longer than expected. Several other studies also reported an experienced lack of information and support after day surgery.[7 8 17] Therefore, this can be considered as an important issue that is still not properly addressed in day-surgical care.

The concept developed by Allvin et al was based on rather old literature and on literature that did not include the patients' perspective.[9] The concept discussed by Lundmark et al was entirely based on interviews with patients.[16] The main dimensions and several of the subdimensions described by Lundmark et al[16] were confirmed in the present study, despite differences in the type of surgery and timing of interviews (which were 12 months postoperative in Lundmark et al.[16] Thus, postoperative recovery after minor and major surgery has similarities, and it is reasonable to discuss whether the postoperative recovery process is generic.

Two crucial central questions for describing postoperative recovery as a concept are: should postoperative recovery be seen an endpoint or as a process,[10] and what does it actually mean to be recovered[11]? Berg et al[7] described postoperative recovery as 'rollback to ordinary life', while Barthelsson et al[22] described it as 'returning to activities of daily life'. Our findings emphasise that postoperative recovery is an individual process, with a clear starting point, the presurgery state, and ending with a new stable state, a state that does not have to be the same as that before surgery. The recovery process is dynamic, and all dimensions are integral and affect each other. Hence, postoperative recovery after day surgery is an individual process. This way of describing the postoperative period as a process has similarities with McMullen et al's[23] study, in which postoperative recovery among patients who had surgery for bladder cancer was described as a transformative process that started with preoperative decision-making and ended with mastery of self-care and reintegration into the activities of daily life. In the present study, it became obvious that the participants were not fully recovered at the time of the interviews—that is, they had not reached a new stable state. Defining the end of the recovery process and determining when the new stable state is established may also depend on the surgery and its consequences for the patient. Patients suffering from a severe disease or injury, as in Lundmark et al's[16] study, did not want to return to their preoperative state, which had included suffering and issues related to decreased lung function

in that case. In another study, patients undergoing major leg amputation due to an arterial disease stated that they experienced recovery after approximately 6 months on regaining their functional independence. Functional independence varied, but was often related to the preoperative level of function.[24] These different perspectives on how to determine when the postoperative process are actually over require further study. For day-surgery patients, follow-up on their recovery after 6–12 months is suggested, along with a focus on their experience of what made them feel recovered. The concept of a new stable state also needs to be further explored and confirmed by other studies.

In present study, the patients used several strategies to cope with symptoms and discomfort in the recovery process, to gradually stabilised in order to achieve a new stable state. Important facilitators for recovery included practical support from next of kin and being-well informed and prepared for the recovery process and it's challenges.[7 8 16 18 22 25 26] Consequently, it is of importance for healthcare personnel provide a person centred as well as systematically follow-up and measure postoperative recovery in order to support and understand the process of postoperative recovery. It is also important to continue to improve preoperative and postoperative information that addresses all the four dimensions, which must be clear, concise and well timed in order to be supportive for day-surgery patients.

## Limitations

This study is a secondary theoretical thematic analysis of interviews that were used in two previous studies. Neither of those sets of interviews were conducted with a focus on the dimensions described in previous concept analyses. This can be seen as both a strength and a limitation, and the chance that interviews focusing on postoperative recovery as a concept may have had a different content cannot be excluded. However, this may also be a strength, as the interviews were not influenced by descriptions of postoperative recovery, unlike those by Allvin *et al*[9] or Lundmark *et al*.[16] It cannot be excluded that additional information would had been sought if further interviews were conducted. However, 38 participants are a rather large sample in qualitative studies, and consistent information emerged from the two samples. Furthermore, the interviews were conducted 14–57 days after the surgery. Therefore, it is possible that there is more information about the recovery process after day surgery that has not been found in this present study. It can also be questioned if theoretical thematic analysis is the most suitable method for analysing our data. As our purpose was to reanalyse the concept of postoperative recovery, and thereby code the data to suite this specific research question, that is, a theoretical interest. Theoretical thematic analysis was, therefore, considered to be the most appropriate method.

## CONCLUSION

This study largely confirmed the concept of postoperative recovery as described by others, and further developed it in a day-surgery context. Postoperative recovery was described as a process with a clear starting point, followed by a dynamic and individual process leading to an experience of a new stable state. The recovery process included physical symptoms, emotions and social and habitual consequences that challenges them. The patients used several strategies to cope with these challenges, and adapt to the recovery process, and gradually stabilised in order to achieve a new stable state.

**Contributors** Study design: UN, MJ and KD. Data collection: KD and KH. Data analysis: UN, MJ, KH, EA and KD. Manuscript writing: UN, MJ, KH, EA and KD.

**Funding** This study was founded by the Swedish Research Council (Vetenskapsrådet), (grant No. 2015–02273).

**Competing interests** None declared.

**Patient and public involvement** Patients and/or the public were not involved in the design, or conduct, or reporting, or dissemination plans of this research.

**Patient consent for publication** Not required.

**Ethics approval** The study follows the Declaration of Helsinki. Samples I and II were approved by the regional ethical review board in Uppsala, Sweden (reference number: 2015/262). Participants received both written and verbal information about the study and gave their written consent.

**Provenance and peer review** Not commissioned; externally peer reviewed.

**Data availability statement** No data are available. No additional data are available.

**ORCID iD**
Ulrica Nilsson http://orcid.org/0000-0001-5403-4183

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
