## [Reviewer comments · BMJ Open]

ARTICLE DETAILS

TITLE (PROVISIONAL)	A journey to a new stable state - further development of the postoperative recovery concept from day surgical perspective: A qualitative study
AUTHORS	Nilsson, Ulrica; Jaensson, Maria; Hugelius, Karin; Arakelian, Erebouni; Dahlberg, Karuna

VERSION 1 – REVIEW

REVIEWER	Julio Fiore Jr. McGill University, Canada
REVIEW RETURNED	04-Apr-2020

GENERAL COMMENTS	Thank you for the opportunity to review the manuscript by Nilsson et al. The manuscript reports a qualitative study aimed to (1) further understand the concept of recovery from the perspective of day-surgery patients and (2) to describe how patients experience postoperative recovery in relation to dimensions and subdimensions of recovery identified in previous literature (Allvin et al. 2007 and Lundmark et al. 2016). To achieve this aim, the authors conducted a secondary analysis of qualitative data obtained in two previous studies (Dahlberg et al. 2018 and Nilsson et al 2019). I believe that this manuscript deals with a topic of great interest to perioperative care clinicians in the current era of patient-centered care, where there is growing interest in understanding surgical recovery from the perspective of patients. My comments and suggestions are described below: 1. The authors should ensure that this manuscript follows current standards for reporting qualitative research studies. I encourage the authors to consult the SRQR recommendations (https://www.equator-network.org/reporting-guidelines/srqr/) and/or the COREQ Checklist (https://www.equator-network.org/reporting-guidelines/coreq/). Ideally, the authors should mention in the methods section which reporting guideline was followed. In addition, I encourage the authors to make a reporting checklist available to readers as an electronic supplement.2. I encourage the authors to further justify their decision to conduct a deductive thematic analysis rather than using a more traditional inductive approach (e.g. grounded theory). The authors anchored their analysis on recovery frameworks that were not specifically focused on day surgery; for example, the study by Lundmark et al (2016) involved patients undergoing lung transplantation (typical hospital length of stay > 1 week) whose experience of recovery can be very different from day surgery patients. For this reason, I wonder if this literature was appropriate to inform the authors' deductive analysis? As recovery frameworks accounting exclusively for the
---

	perspective of day surgery patients are currently inexistent, I wonder if an inductive approach would be more appropriate (and valuable)? 3. The authors should consider specifying the sampling strategy used to identify participants: How were participants selected (e.g. purposive, convenience, consecutive, snowball sampling)? How were participants approached? Most importantly, did the original studies that provided data for this secondary analysis use the same sampling strategy? Any risk of selection bias should be addressed by the authors. 4. Saturation is a core principle used in qualitative research to determine when enough data were collected to develop a robust and valid understanding of the phenomenon being studied. The authors should consider reporting how and when they decided that no further interviews were necessary. Is the concept of saturation relevant to their deductive analysis approach? If not, why? As this is a secondary analysis of previous collected data, the authors may have to refer to the primary studies when reporting on the adequacy of their sample size. 5. I encourage the authors to report more details about the interview guides used in this study. Was the same interview guide used for both patient samples? Ideally, the interview guide(s) should be made available to readers as an electronic supplement. 6. In the discussion, I encourage the authors to further explore the value of this study for perioperative care clinicians and researchers. How can the study results be used to improve patient care and/or inform future research? 7. I encourage the authors to revisit their conclusion (in both the abstract and main manuscript). Some of the statements made are not justified by the study results, e.g., "... it is important for healthcare providers to continue to improve preoperative and postoperative information, which must be clear, concise and well timed in order to be supportive for day-surgery patients." I would be pleased to review a new version of this manuscript if the authors believe that their study is suitable for reporting according to current guidelines. The potential methodological limitations here described should also be addressed by the authors. Otherwise, I regrettably support that this manuscript should be rejected.
--	---

REVIEWER	Professor Anna Forsberg Institute of Health Sciences at Lund University, Sweden
REVIEW RETURNED	06-Apr-2020

GENERAL COMMENTS	I congratulate you to a well written paper moving the knowledge forward regarding postoperative recovery.. The introduction is relevant and the rational is clear with an appropriate method to answer the research question. I agree with the crucial question if recovery should be viewed as process or an end-Point. Your findings suggest in a reasonable way that it is more of a process. I have two comments, one very minor and one that might need some more work:  1. Please change the word adaption to adaptation. 2. It would be useful to expand the findings also to include a description of the meaning of being in a stable state. You describe it shortly in the discussion but my impression is that you have enough
--

	data to also present an interpretation what it actually means to be in a stable state. That would be the inductive part of the re-analysis which is no problem from a methodological perspective. You started deductively and end up with an inductive interpretation of the crucial finding of reaching a stable state, thus perhaps constituting the end-point of the process of recovery from the patients' perspective.
--	--

VERSION 1 – AUTHOR RESPONSE

Reviewer: 1

Reviewer Name: Julio Fiore Jr.

Institution and Country: McGill University, Canada Please state any competing interests or state 'None declared': None declared

Please leave your comments for the authors below Thank you for the opportunity to review the manuscript by Nilsson et al. The manuscript reports a qualitative study aimed to (1) further understand the concept of recovery from the perspective of day-surgery patients and (2) to describe how patients experience postoperative recovery in relation to dimensions and subdimensions of recovery identified in previous literature (Allvin et al. 2007 and Lundmark et al. 2016). To achieve this aim, the authors conducted a secondary analysis of qualitative data obtained in two previous studies (Dahlberg et al. 2018 and Nilsson et al 2019). I believe that this manuscript deals with a topic of great interest to perioperative care clinicians in the current era of patient-centered care, where there is growing interest in understanding surgical recovery from the perspective of patients. My comments and suggestions are described below:

1. The authors should ensure that this manuscript follows current standards for reporting qualitative research studies. I encourage the authors to consult the SRQR recommendations (<https://eur01.safelinks.protection.outlook.com/?url=https%3A%2F%2Fwww.equator-network.org%2Freporting-guidelines%2Fsrqr%2F&data=02%7C01%7Culrica.nilsson%40ki.se%7C81afd434391a48d0640208d7ff2cb90c%7Cbff7eef1cf4b4f32be3da1dda043c05d%7C0%7C0%7C637258439860482336&sdata=TicxQeOzhVP55TQkBTpNtoYqPcJ6ZGIOgagX9pGuDTo%3D&reserved=0>) and/or the COREQ Checklist (<https://eur01.safelinks.protection.outlook.com/?url=https%3A%2F%2Fwww.equator-network.org%2Freporting-guidelines%2Fcoreq%2F&data=02%7C01%7Culrica.nilsson%40ki.se%7C81afd434391a48d0640208d7ff2cb90c%7Cbff7eef1cf4b4f32be3da1dda043c05d%7C0%7C0%7C637258439860482336&sdata=JyrrylKJaxtTWqolZ0tF%2FfOx%2BEk8uL9OyAor5eAlhuA%3D&reserved=0>). Ideally, the authors should mention in the methods section which reporting guideline was followed. In addition, I encourage the authors to make a reporting checklist available to readers as an electronic supplement.

Both SRQR and COREQ checklist are submitted.

2a. I encourage the authors to further justify their decision to conduct a deductive thematic analysis rather than using a more traditional inductive approach (e.g. grounded theory). The authors anchored their analysis on recovery frameworks that were not specifically focused on day surgery; for example, the study by Lundmark et al (2016) involved patients undergoing lung transplantation (typical hospital length of stay > 1 week) whose experience of recovery can be very different from day surgery patients. For this reason, I wonder if this literature was appropriate to inform the authors' deductive analysis?

In the background we have clarified that Lundmark et al study is a further development of Allvin et als work. As we have performed inductive analysis earlier in order to describe patients experience of their postoperative recovery, we now identified a need for a further development of the concept and thought that it would be suitable to use Allvin et al and Lundmark et als dimensions and sub-dimensions as a framework. We also discuss similarities and differences between our

results and Alvin et al and Lundmark et al, such as the differences in the type of surgery that the participants had undergone.

2b. As recovery frameworks accounting exclusively for the perspective of day surgery patients are currently inexistent, I wonder if an inductive approach would be more appropriate (and valuable)?

Thank you for your valuable input. This comment has made us further reflect of our methodological choices made in the beginning of this process. We do not agree that an inductive analysis would be more appropriate however after going back to the methodological literature referred to in the analysis, we realize that the concept theoretical thematic analysis is more appropriate to use than deductive thematic analysis. *“Theoretical thematic analysis would tend to be driven by the researcher’s theoretical or analytic interest in the area and thus more explicit analyst-driven..... The choice between inductive and theoretical maps onto how and why you are coding the data well. You can either code for a quit specific research question (which maps onto the more theoretical approach) or the specific research question through the coding process can evolve through the coding process (which maps onto inductive analysis) ”* Page 12 in Braun V, Clarke V. Using thematic analysis in psychology. Qualitative research in psychology. 2006;3(2):77-101 The analys method is also diskussed in the Limitaion section.

3. The authors should consider specifying the sampling strategy used to identify participants: How were participants selected (e.g. purposive, convenience, consecutive, snowball sampling)? How were participants approached? Most importantly, did the original studies that provided data for this secondary analysis use the same sampling strategy? Any risk of selection bias should be addressed by the authors.

Information about sampling has been added to the Methods section. The concept of selection bias is not used in purposive sampling, therefore we have not addressed this in the paper.

4. Saturation is a core principle used in qualitative research to determine when enough data were collected to develop a robust and valid understanding of the phenomenon being studied. The authors should consider reporting how and when they decided that no further interviews were necessary. Is the concept of saturation relevant to their deductive analysis approach? If not, why? As this is a secondary analysis of previous collected data, the authors may have to refer to the primary studies when reporting on the adequacy of their sample size.

Information about data saturation and sample size has been included in the methods section and in the limitation section.

5. I encourage the authors to report more details about the interview guides used in this study. Was the same interview guide used for both patient samples? Ideally, the interview guide(s) should be made available to readers as an electronic supplement.

Amended, an additional file has been submitted.

6. In the discussion, I encourage the authors to further explore the value of this study for perioperative care clinicians and researchers. How can the study results be used to improve patient care and/or inform future research?

Information about the importance of support, follow-up and to assess postoperative recovery have been inserted.

7. I encourage the authors to revisit their conclusion (in both the abstract and main manuscript). Some of the statements made are not justified by the study results, e.g., “... it is important for healthcare providers to continue to improve preoperative and postoperative information, which must be clear, concise and well timed in order to be supportive for day-surgery patients.”

The last two sentences have been deleted from the conclusion and some parts has also been re-written.

I would be pleased to review a new version of this manuscript if the authors believe that their study is suitable for reporting according to current guidelines. The potential methodological limitations here described should also be addressed by the authors. Otherwise, I regrettably support that this manuscript should be rejected.

Reviewer: 2

Reviewer Name: Professor Anna Forsberg

Institution and Country: Institute of Health Sciences at Lund University, Sweden Please state any competing interests or state 'None declared': None declared

Please leave your comments for the authors below Dear Authors, I congratulate you to a well written paper moving the knowledge forward regarding postoperative recovery.. The introduction is relevant and the rational is clear with an appropriate method to answer the research question. I agree with the crucial question if recovery should be viewed as process or an end-Point. Your findings suggest in a reasonable way that it is more of a process. I have two comments, one very minor and one that might need some more work:

1. Please change the word adaption to adaptation.

Changed as requested

2. It would be useful to expand the findings also to include a description of the meaning of being in a stable state. You describe it shortly in the discussion but my impression is that you have enough data to also present an interpretation what it actually means to be in a stable state. That would be the inductive part of the re-analysis which is no problem from a methodological perspective. You started deductively and end up with an inductive interpretation of the crucial finding of reaching a stable state, thus perhaps constituting the end-point of the process of recovery from the patients' perspective.

Thank you for your suggestion, we have rearranged the manuscript so that the inductive interpretation of the recovery process is included in the Findings section.

VERSION 2 – REVIEW

REVIEWER	Julio F Fiore Junior McGill University, Canada
REVIEW RETURNED	12-Jun-2020

GENERAL COMMENTS	Thank you for the opportunity to provide further input to the manuscript by Nilsson et al. The manuscript has been thoroughly revised and much improved after the reviewers' initial feedback. I have a few further comments and suggestions, as described below:  1. In the abstract, the conclusion that "It is therefore important to systematically follow-up and measure all four dimensions of postoperative recovery in order to support and understand the process of postoperative recovery" is not justified by the study results. I encourage the authors to exclude or rephrase this statement (making it less affirmative). 2. In the results section, the authors included an interesting, but very theoretical, paragraph describing the process of recovery ending in a 'new stable state'. This paragraph presents an interpretation of the study results, rather than a result itself. Therefore, this paragraph would probably fit better in the discussion section. 3. I could not find Figure 1 in the files submitted to the journal.
---

VERSION 2 – AUTHOR RESPONSE

1. In the abstract, the conclusion that “It is therefore important to systematically follow-up and measure all four dimensions of postoperative recovery in order to support and understand the process of postoperative recovery” is not justified by the study results. I encourage the authors to exclude or rephrase this statement (making it less affirmative).

The sentence has been rephrased: “To follow-up and measure all four dimensions of postoperative recovery in order to support and understand the process of postoperative recovery is therefore recommended”.

2. In the results section, the authors included an interesting, but very theoretical, paragraph describing the process of recovery ending in a ‘new stable state’. This paragraph presents an interpretation of the study results, rather than a result itself. Therefore, this paragraph would probably fit better in the discussion section.

In previous review of our manuscript the Reviewer Professor Anna Forsberg suggested that “*It would be useful to expand the findings also to include a description of the meaning of being in a stable state*” We thought this was an excellent idea and followed her suggestion and prefer to not change this. The theory “new stable state” is also reflected upon in the discussion section.

3. I could not find Figure 1 in the files submitted to the journal.

Figure 1 was uploaded and could be found on page 56 of 65 in the version that we submitted 31 Maj 2020.